# Identification and Functional Analysis of *MAPKAPK2* in *Hyriopsis cumingii*

**DOI:** 10.3390/genes13112060

**Published:** 2022-11-07

**Authors:** Yang Gu, Meiling Liu, Yayu Wang, Yingduo Huo, Zongyu Liu, Wu Jin, Guiling Wang

**Affiliations:** 1Key Laboratory of Freshwater Aquatic Genetic Resources, Ministry of Agriculture and Rural Affairs, Shanghai Ocean University, Shanghai 201306, China; 2National Demonstration Center for Experimental Fisheries Science Education, Shanghai Ocean University, Shanghai 201306, China; 3Shanghai Engineering Research Center of Aquaculture, Shanghai Ocean University, Shanghai 201306, China; 4Key Laboratory of Integrated Rice-Fish Farming Ecology, Ministry of Agriculture and Rural Affairs, Freshwater Fisheries Research Center, Chinese Academy of Fishery Sciences, Wuxi 214081, China; 5Wuxi Fisheries College, Nanjing Agricultural University, Wuxi 214128, China

**Keywords:** *Hyriopsis cumingii*, *MK2*, reproductive cells development, immune stimulation

## Abstract

*MAPKAPK2* (*MK2*) is an important regulator of the p38 mitogen-activated protein kinase (p38 MAPK) pathway, which is involved in a plethora of cellular processes concluding the development of gamete cells in meiosis and resisting pathogenic bacterial infestation. *Hyriopsis cumingii* is a significant mussel resource in China and a good material for pearl breeding. To explore the role of *MK2* in *H. cumingii, MK2* was identified and cloned, whose full-length cDNA was 1568 bp, including 87 bp in 5′ UTR, 398 bp in 3′ UTR, and 1083 bp in the open reading frame (ORF) region, encoding 360 amino acids. The expression of *MK2* was the highest in the gills. Meanwhile, there was a significant difference in the gonads. After *Aeromonas hydrophila* and *Lipopolysaccharide* (LPS) infestation, the transcript level of the *MK2* was upregulated in the gills. It indicated that *MK2* might be involved in the innate immune response of *H. cumingii* after a pathogenic attack. After quantifying *H. cumingii* of different ages, it was found that the expression of *MK2* was highest at 1 year old. In situ hybridization (ISH) results showed that the blue-purple hybridization signal was very significant in the oocytes and egg membranes of the female gonads of *H. cumingii*. The expression of *MK2* increased gradually at the age of 1 to 5 months and showed a downward trend at the age of 5 to 8 months. It was suggested that *MK2* might play an important role in the formation of primitive germ cells in *H. cumingii*. To sum up, *MK2* might not only be involved in the immune response against pathogenic bacterial infection but also might play an important role in the development of the gonads in *H. cumingii*.

## 1. Introduction

MAPK plays an important role in stabilizing and facilitating pole and chromosome separation. p38 MAPK is one of the three major members of the MAPK family that regulates cellular responses, including cell proliferation and immune response, as well as cell growth and differentiation [1]. *MK2* is one of the downstream substrates of p38 MAPK [2]. Studies have shown that *MK2* is involved in the division and development of germ cells. In porcine oocytes, *MK2*, localizing at the plus end of spindle microtubules, is a critical regulator of meiotic cell cycle progression [3]. Previous studies have elucidated a novel role of MK2 in Drosophila spermatogenesis that MK2 phosphorylates the RNA-binding protein Dazl to regulate spermatogenesis [4]. In the yolk cell, activated MK2 could regulate the process of the epiblast. In zebrafish, the p38 MK2 kinase cascade regulates F-actin activity around the yolk cell rim and modulates actin contraction at the blastoderm margin, causing the embryonic pore to gradually close during blastoderm development [5]. Experimental results demonstrate that MK2 plays a sex-specific role in mouse osteoclastogenesis. MK2 signaling is critical for prefusion genes only in male osteoblasts [6].

Besides its role in the development of germ cells, *MK2* is also described as a mediator of p38-driven signaling and is commonly seen in inflammatory responses, enhancing the transcription levels of inflammatory cytokines [7,8,9,10,11]. Stimulation by pathogens and substances of their origin, such as CpG dinucleotide-rich bacterial DNA and bacterial LPS, activates the innate immune response, in which MAPK is critically involved [12]. MK2-deficient mice displayed diminished resistance to Listeria monocytogenes on account of impaired control of bacterial growth [13]. Activation of MK2 has been found to be a key driver of intestinal inflammation in patients with *Clostridium difficile* infection in infected animals and humans [14]. *A. hydrophila*, widely found in freshwater, is a common pathogenic bacterium that causes aquaculture diseases and a classic “human-animal-fishery” zoonotic pathogen [15]. *A. hydrophila* can produce highly toxic exotoxins which are critical components of bacterial pathogenesis. Usually, it infects the intestine, causing disease in aquatic organisms [16,17,18]. Mortality caused by *A. hydrophila* is as high as 65-90% in the cultivation of *H. cumingii* [19].

*H. cumingii* is a unique mussel resource in China and is a good material for pearl breeding [20]. Meanwhile, the pearl production performance of male mussels is superior to that of female mussels [21]. Thus, achieving a monosexual culture or improving breeding capacity will drive the development of the pearl culture industry. On account of sex chromosome deletion, the regulatory role of sex-related genes has become critical to research. Meanwhile, in the process of pearl culture, insert transplantation is the core part of the artificial pearl cultivation process. However, after transplantation, the cultured pearl shell will generally have an immune rejection to the small pieces of exogenous mantle cells, and at the same time, the organism is also harmed by pathogenic bacteria in the external environment, resulting in inflammatory reactions and even death [22]. This article aims to explore the role of *MK2* in *H. cumingii* and provide a scientific and theoretical basis to promote the efficient and sustainable development of *H. cumingii* cultures.

## 2. Materials and Methods

### 2.1. Experimental Animals, Preparation and Sample Collection

All *H. cumingii* individuals used were taken from the Wuyi experimental farm in Jinhua, Zhejiang Province. The selected individuals were then transported back to Shanghai Ocean University, where they were used for subsequent experiments after a week of temporary rearing in the laboratory. The study followed the guidelines of the Institutional Animal Care and Use Committee (IACUC) of Shanghai Ocean University, China. Gonadal tissue was collected from *H. cumingii* aged 1 to 8 months. Various tissues (liver, gill, obturator, mantle, foot, and gonads) were removed from three healthy males and three sexually mature females of *H. cumingii*. The gonads of 1-year-old *H. cumingii* were taken and paraffin sections were made for subsequent experiments.

### 2.2. Immune Response

Healthy and similar-sized 1-year-old male *H. cumingii* were selected and temporarily raised at 22 ± 4 °C for a week. At the end of the temporary care, they were divided into three groups. One group was set as a blank control, and the remaining two groups were used for *A. hydrophila* and LPS infestation, respectively. The concentration of *A. hydrophila* was made up to 10^9^ CFU/mL using PBS buffer solution [22]. Each mussel was injected with a 1 mL syringe with *A. hydrophila* and LPS (1 mg/mL), 50 μL of each in the adductor, and the injected mussels were returned to the same environment for temporary breeding. Gills were collected at 0 h, 3 h, 6 h, 12 h, 24 h, 36 h, 48 h and 96 h after treatment. All tissue samples were frozen in liquid nitrogen immediately after being removed and subsequently stored at −80 °C for subsequent RNA extraction. The samples used for in situ tissue hybridization were gonadal tissues of 1-year-old *H. cumingii*.

### 2.3. RNA Extraction and cDNA Synthesis

Total RNA was extracted using Trizol reagent immediately after the removal of tissue samples from −80 °C. The entire process was performed on ice. The quality and concentration of the extracted RNA were detected using 1% agarose gel electrophoresis (180 V, 200 mA) and NanoDrop 2000. The RNA was reverse transcribed (PrimeScript^TM^ RT Reagent Kit with gDNA Eraser, TaKaRa, Dalian, Chian) to obtain a cDNA template [23]. After electrophoretic detection of the PCR-amplified cDNA products, containing target-striped adhesive blocks were cut, recovered and purified, and the purified cDNA was cloned to PMD19-T (TaKaRa, Dalian, Chian) as a suitable vector. The ligation product was transformed to DH5α (TaKaRa, Dalian, Chian) for sequencing.

### 2.4. MK2 Full-Length Acquisition and Sequence Analysis

The partial sequences of *MK2* obtained from the transcriptome library [24] were used to design the inner and outer primers for 3′ RACE (Table 1), and the full-length amplification of the 3′ end of the *MK2* was performed by referring to the instructions of the SMART 3′ RACE Kit (Clontech, America). The sequence data has been uploaded to the NCBI. The GenBank accession number is OP377074, obtained on 7 September 2022. The ORF finder (https://www.ncbi.nlm.nih.gov/orffinder/, accessed on 28 October 2022) and BLAST sequence of nucleotides (https://blast.ncbi.nlm.nih.gov/Blast.cgi, accessed on 28 October 2022) were used to obtain the ORF forecast and analysis of the amino acid sequence column homology. Primers were designed using Primer Premier 5.0 software. ProtParam (https://web.expasy.org/protparam/, accessed on 28 October 2022) was used to analyze the basic physical and chemical properties. SignalP 4.1 program (http://www.cbs.dtu.dk/services/SignalP/, accessed on 28 October 2022) was used to predict genes in the existence of a signal peptide. The SMART program (http://smart.embl-heidelberg.de/, accessed on 28 October 2022) was used to predict the protein domain. The SWISS-MODEL program (https://swissmodel.expasy.org/, accessed on 28 October 2022) was used to predict the tertiary structure. TMHMM Server v2.0 program (http://www.cbs.dtu.dk/services/TMHMM/, accessed on 28 October 2022) was used to predict the membrane structure. The phylogenetic tree was constructed using the ClustalW2 program and Neighbor-Joining (NJ) method in MEGA 7.0, and the reliability of the phylogenetic tree was evaluated by 1000 times bootstrap.

### 2.5. Quantitative Real-Time PCR

*EF-1α* was selected as the internal reference gene, and the expression of *MK2* was detected using the CFX96 Touch Real-Time PCR Detection System (Bio-Rad, Shanghai, China). The reaction system (20 μL) was as follows: cDNA 1.6 μL, 2 × TB Green 10 μL, RNase-free water 6.8 μL, upstream and downstream primers 0.8 μL each. Each sample had three replicates, and the primers used are shown in Table 1. The experimental procedure was as follows: 95 °C for 3 min; 95 °C for 5 s, 60 °C for 30 s, 95 °C for 10 s, 40 cycles in total. The relative expression was calculated according to the 2^−ΔΔCT^ method. Significant differences were analyzed using SPSS23.0 software and plotted by Prism8.0.

### 2.6. In Situ Hybridization

Primers used for in situ hybridization are shown in Table 1. The gonadal cDNA of *H. cumingii* was amplified as a template, and the target strips were recovered, purified and used as a template to obtain labeled probes using a T7 High Efficiency Transcription Kit (TransGen, Beijing, China) and DIG RNA Labeling Mix (Roche Applied Science, Basel, Switzerland), and the resulting probes were purified and stored at −80 °C. Paraffin sections of the gonads of 1-year-old healthy, similar-sized males and females were taken for in situ hybridization using a DIG Nucleic Acid Detection Kit (Boster, California, USA), and the hybridization signal was observed under a microscope (Leica, Wetzlar, Germany) and photographed.

### 2.7. Data Processing, Statistical Analysis and Graph Production

The independent samples *t*-test and one-way analysis of variance ANOVA in the SPSS23.0 software were used to analyze the significant differences among the data groups. When *p* < 0.05, a significant difference was considered. Prism 8.0 software was used to produce bar graphs and line graphs. All values in this article are expressed as mean ± standard deviation (SD) in the graphs.

## 3. Result

### 3.1. Full-Length Cloning and Sequence Characterization of MK2 cDNA in H. cumingii

The sequence of *MK2* cDNA was cloned, with a total length of 1568 bp, including 87 bp in 5′UTR, 398 bp in 3′UTR and 1083 bp (88-1170) in the ORF region, encoding 360 amino acids (Figure 1). It contained 74 acidic amino acids and 54 basic amino acids. The predicted relative molecular mass was 40.87, the isoelectric point was 7.56, and the average hydrophilic coefficient was −0.361, which was presumed to be a hydrophilic protein. The protein does not possess a transmembrane structure or signal peptide and belongs to an intracellular protein. There was a protein kinase structural domain at amino acids 22-283 of the MK2 protein, which was predicted by HUMER; SMART and conserved domains analysis showed that this was the stKC-Mapkapk structural domain. The protein tertiary structure of *MK2* was predicted by I-TASSER (Figure 2).

BlastP showed that *MK2* was 70.59~80.13% similar to that in other species. For instance, the similarity of the *MK2* amino acid sequences in *Crassostrea virginica* (XP_022299678.1), *Mytilus edulis* (CAG2253780.1) and *Lingula Anatina* (XP_013393146.1) were 82.06%, 81.62% and 76.32%, respectively. The sequence comparative analysis showed that *Mauremys Reevesii* (XP_039390375.1), *Phascolarctos Cinereus* (XP_020858778.1), and *Neopelma Chrysocephalum* (XP_027550838.1) had nearly 40 more amino acid residues than *H. cumingii*, with proline accounting for the majority. The phylogenetic tree showed that *H. cumingii* and other bivalve species such as *Crassostrea virginica* (XP_022299678.1) and *Mytilus coruscus* (CAC5402940.1) were clustered on one branch, gastropods and vertebrates were clustered on another branch (Figure 3).

### 3.2. Expression of MK2 in Various Tissues of the Mature H. cumingii

*MK2* was expressed in all tissues of males and females, and there were significant differences. In males, *MK2* was highly expressed in the gills, followed by the gonads, and lowest in the mantle. In female mussels, *MK2* expression was highest in the gonads and lowest in the liver. The expression of *MK2* was significantly higher in females than in males in the adductor, foot, and gonads, and was significantly lower in females in the liver and gills compared with males. (Figure 4).

### 3.3. MK2 Responded to the Transcriptional Level after Infection by A. hydrophila and LPS in H. cumingii

The expression of *MK2* gradually augmented with the time of immune response, within 48 h after injection of *A. hydrophila* and LPS. There was a decreasing trend from 48 h to 96 h, but the expression was still higher than that during 0–24 h (Figure 5). It emerged that the transcript level of *MK2* increased faster in 24–48h after *A. hydrophila* injection while in 12–48 h after LPS injection.

### 3.4. Expression of MK2 in Juvenile H. cumingii

During early gonadal development in *H. cumingii* (from the fertilized egg to 8 months old), the expression of *MK2* increased gradually from the age of 1 to 5 months, reaching the highest expression level at the age of 5 months. The transcription level showed a downward trend from 5 to 8 months. The expression of *MK2* was significantly higher at the age of 4 to 5 months old than that in other months-old mussels (Figure 6).

### 3.5. Expression of MK2 in Adult H. cumingii

The expression of *MK2* decreased gradually at the age of 1 to 3 years, reaching the highest expression level at the age of 1 year. There was a highly significant difference in *MK2* expression between male and female individuals at 1 and 2 years old, and no significant difference at the age of 3 years.

### 3.6. Localization of MK2 in H. cumingii

ISH was used to detect the localization of *MK2* in the gonad (Figure 7). The results showed that the blue-purple hybridization signal was very significant in the oocytes and egg membranes of the female gonads in the *H. cumingii* experimental group. There was a small number of blue-purple hybridization signals in the spermatogonia, spermatocytes and follicle walls of the male gonads in the experimental group, while there was no hybridization signal in the female and male-negative control groups.

## 4. Discussion

In this study, *MK2* was identified and cloned from *H. cumingii*. MK2 is one of the downstream protein kinase subfamilies of MAPK, which mediates a variety of important cellular physiological responses [7]. The N-terminus of *MK2* in mammals contains a proline-rich region, but there is none in *H. cumingii*. The series of comparisons showed that vertebrates have nearly 40 more amino acid residues than bivalves such as *H. cumingii*, and proline occupies the majority of the residues (Figure 3). It could be on account that the biological functions of MK2 protein became more abundant with biological evolution. The full length of *H. cumingii MK2* cDNA was 1568 bp, encoding 360 amino acids (Figure 1). There was an stKC-Mapkapk protein kinase structural domain at amino acid positions 22-283 (Figure 2). The results of multiple sequence alignment analysis showed that the protein kinase structural domain was highly conserved among species, appearing that MK2 had similar functions during the evolution of organisms. The results of the phylogenetic tree presented that *H. cumingii* is more closely related to other bivalve shellfish, in general agreement with the results of the multiple sequences (Figure 3).

*MK2* was more highly expressed in the gills of *H. cumingii*, compared with other tissues (Figure 4). Bivalves, as invertebrates, depend merely on the innate immune response [25,26]. Gills play an important role in the innate immune response. Hemolymph cells flow out of the arteries, pass through all body organs, and return to the heart through respiratory structures (gills) and the venous sinuses. Therefore, the gills are the main body organ participating in the immune response [27]. Under the stimulation of foreign body infection, the MAPK signaling pathway is activated in *Cyprinus carpio, Ctenopharyngodon idella* and *Procambarus clarkii* [28,29,30], which affects the transcription levels of some genes related to inflammation in immune-related tissues (gill and hepatopancreas). For instance, exposure to *Chlorpyrifos* (an organophosphate insecticide) activated the MAPK pathway, promoted inflammatory damage and induced necrotic apoptosis in carp gills [30]. MK2 is one of the downstream substrates of MAPK. The high expression of *MK2* in the gills of *H. cumingii* implies that it may play a vital part in the resistance to pathogenic invasion (Figure 4). Studies on the involvement of *MK2* in the innate immune response of bivalve shellfish are mostly seen in post-insertional transplantation responses in pearl mussels. It has been demonstrated that the MAPK signaling pathway, as an important immune signaling pathway, plays a crucial part in the immune response after xenotransplantation surgery in the freshwater pearl mussel, *H. cumingii* [31], and the marine shell mussel, *Pinctada martensii*. After the post-insertional transplantation of pearl mussels, it is easy to receive infection with pathogenic bacteria, resulting in death. LPS is a component of the cell wall of gram-negative bacillus and can be used as an immune enhancer for nonspecific immune aquatic animals. Additionally, *A. hydrophila* is the main pathogenic bacteria in the breeding process of the mussels. LPS and *A. hydrophila* were used to infect *H. cumingii*. It was found that the expression level of *MK2* gradually increased during 0-36 h and gradually decreased during the next 60 h. After LPS injection, transcript levels were consistently elevated during 0–48 h (Figure 5). According to this result change, it indicated that *MK2* received the effect of the *A. hydrophila* attack. Although the expression of *MK2* was down-regulated throughout 48-96 h, it still maintained a level compared to that during 0–3 h (Figure 5). Therefore, the comprehensive analysis showed that *MK2* had an important role in participating in the innate immunity to effectively recognize pathogenic bacteria and defend against a bacterial attack in *H. cumingii*.

The results of *MK2* quantification in various tissues of *H. cumingii* showed that there were significant differences between males and females, except for the mantle (Figure 4). The expression level was significantly higher in the ovary than that in the testis (*p* < 0.05). Studies have revealed in the past that the MAPK cascade is involved in regulating the meiotic cell cycle progression of oocytes [32,33], during meiosis in mammalian oocytes, especially spindle assembly and microtubule organization during meiosis in mammalian oocytes [34,35,36]. *MK2* can be involved in regulating spermatogenesis [37], in addition, *MK2*, as a significant regulator, is also involved in the meiosis and maturation of porcine oocytes [3]. The effect of *MK2* in mouse oocytes resembles that in porcine oocytes, although the distribution pattern and mode of action are different. The expression of *MK2* was significantly higher at 4–5 months old than that at other ages during the gonadal development of juveniles (1 to 8 months old) of *H. cumingii* (Figure 6). The gonadal tissues started to appear in the visceral mass at 5 months old, while the primordial germ cells of *H. cumingii* started at 4 months old [37]. For *H. cumingii*, sex differentiation is complete at 1 year. At 2 years old, gonads begin to mature, and it achieves fully mature gonadal development at 3 years old. Given that the *MK2* expression was the highest in the gonads of 1-year-old *H. cumingii* (Figure 7), the gonadal tissue of 1-year-old *H. cumingii* was selected for in situ hybridization experiments. The results of the ISH showed that there were obvious blue-purple hybridization signals in the oocytes and egg membranes of the female gonads of *H. cumingii* (Figure 8). It could be assumed that *MK2* plays a role in the development of the ovaries. Considering its high expression at 4 to 5 months old, it was hypothesized that *MK2* might play an important role in the formation of primitive germ cells of *H. cumingii*.

In conclusion, this study showed that *MK2* can be involved in an innate immune response after a pathogenic attack and the formation of primitive germ cells in *H. cumingii*. Exploring the role of *MK2* in *H. cumingii* provides not only the basic foundation for enhancing the resistance of the organism to pathogenic bacteria but also a fundamental reference for screening genes related to sex determination. Because of the deletion of dysmorphic chromosomes in *H. cumingii*, genes related to sex can be a key for studying the molecular mechanisms of sex determination. Grasping the sex-determination mechanism of *H. cumingii* may provide references for monosexual pearl culture. However, whether *MK2* is involved in sex determination requires further study.

## 5. Conclusions

In summary, the role of *MK2* in *H. cumingii* was explored in this study. *MK2* can play an important role in the innate immune response and gonadal development of *H. cumingii*. The full-length *MK2* was identified and cloned for the first time in this experiment. Sequence analysis showed that the *MK2* was 1568 bp long, encoding 360 amino acids, and contained a highly conserved stKC-Mapkapk structural domain. The *MK2* was expressed in all tissues of the mussel, with higher expression in the gills and gonads. The results of the immune response and gill quantification indicated that *MK2* was involved in the innate immune response of *H. cumingii* after a pathogenic attack. The results of in situ hybridization and gonad quantification implied that *MK2* might be involved in the gonadal development and play an important role in the formation of primitive germ cells of *H. cumingii*, which will provide a fundamental reference for screening genes related to sex determination.

## Figures and Tables

**Figure 1 genes-13-02060-f001:**
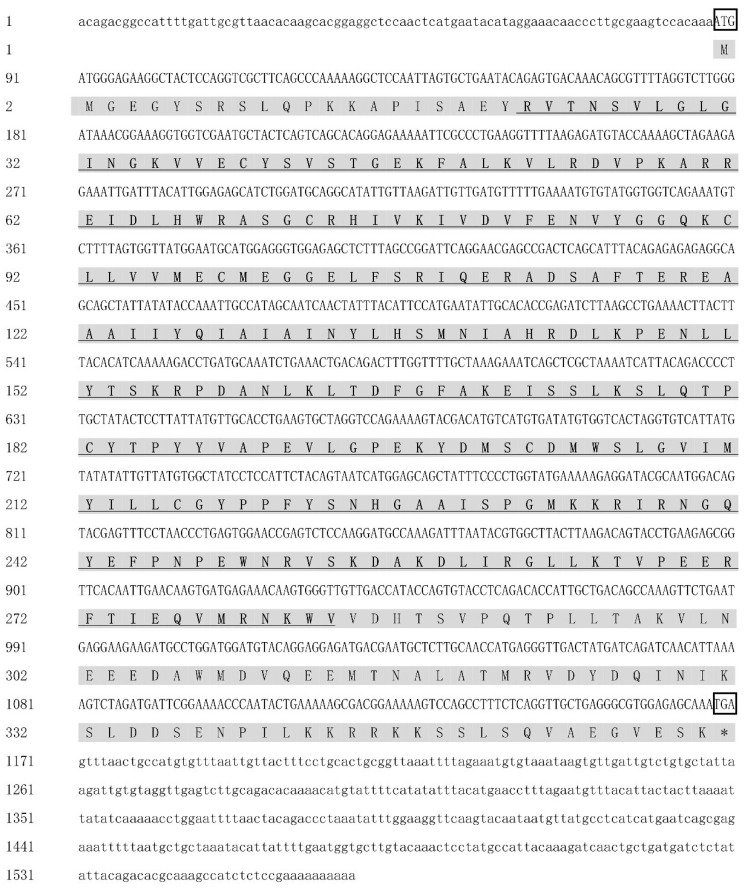
The full-length *MK2* sequence and amino acid encoding in *H. cumingii*. The “ATG” and “TGA” inside the box are the initiation codon and termination codon. The shaded parts are the amino acids encoded by an open reading frame, and the area highlighted in bold and underlined is the predicted domain.

**Figure 2 genes-13-02060-f002:**
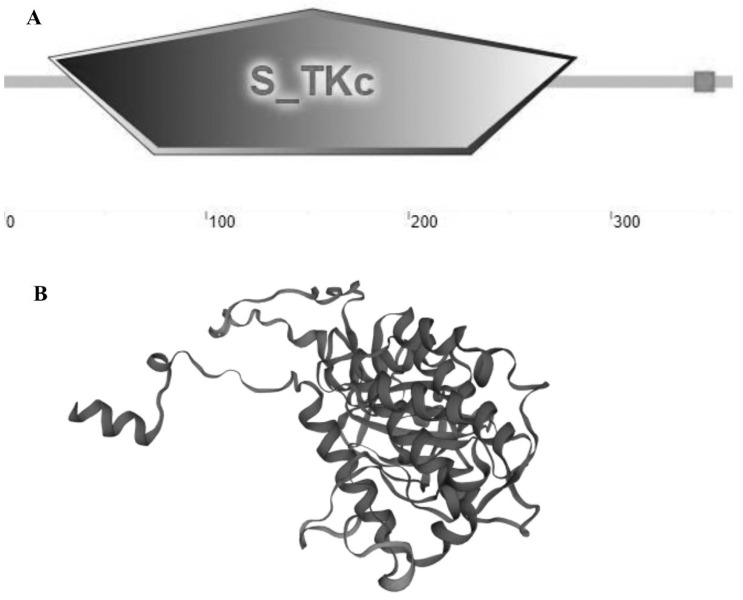
Structural prediction analysis of the MK2 protein of *H. cumingii*. (**A**) SMART and Conserved Domains showed that the domain was the stKC-Mapkapk domain. (**B**) Prediction of the tertiary structure of MK2 protein.

**Figure 3 genes-13-02060-f003:**
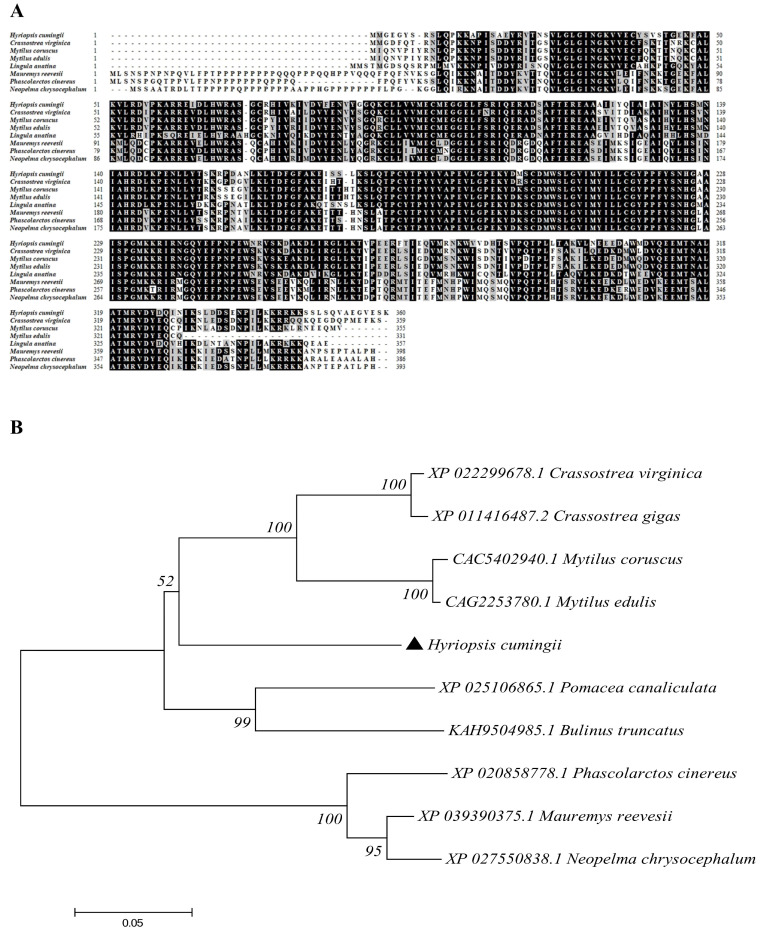
Sequence alignment and phylogenetic tree construction of the *MK2* amino acid sequences in *H. cumingii* with other species. (**A**) Multiple alignment analysis of the *MK2* amino acid sequences of *H. cumingii* with those in other species. The *MK2* amino acids of the selected species are enumerated. The black parts are amino acid residues conserved and the gray parts are similar to amino acid residues in different species. (**B**) NJ Phylogenetic tree of MK2 protein of *H. cumingii* and other creatures. The phylogenetic tree was constructed and inferred by maximum likelihood analysis using MEGA7.0. The numbers are the bootstrap test confidence values with 1000 bootstrap repeats.

**Figure 4 genes-13-02060-f004:**
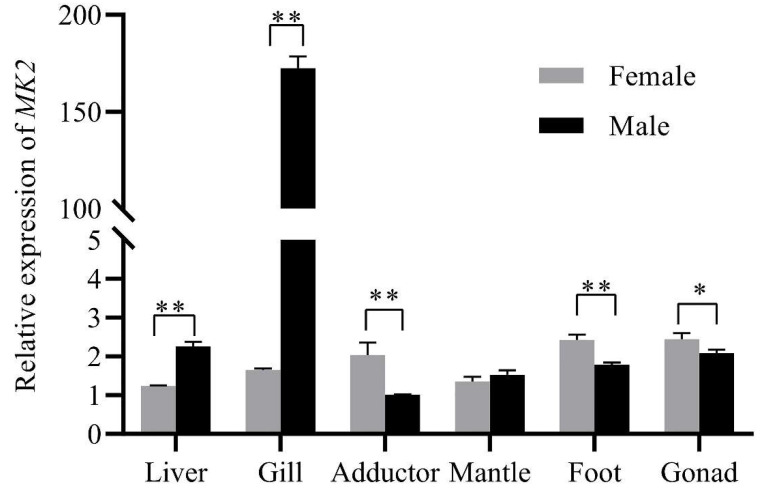
Relative expression of *MK2* in different somatic tissues in female and male *H. cumingii*. *EF1α* was selected as the internal reference gene, and the quantitative results all had three sets of biological replicates. The horizontal axis of the coordinates was the different tissues of *H. cumingii*, and the vertical axis is the relative expression of *MK2*. The analysis of significance was marked with an asterisk. A single asterisk means there is a significant difference (* *p* < 0.05), and double asterisks mean a highly significant difference (** *p* < 0.01).

**Figure 5 genes-13-02060-f005:**
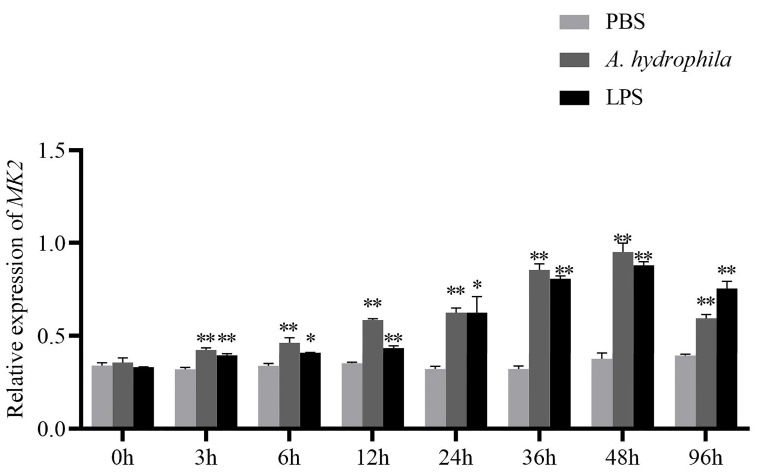
Relative expression of *MK2* in the gills of *H. cumingii* after *A. hydrophila* and LPS infection. PBS buffer solution was injected into the control group. The horizontal axis was the time of immune response after injection. mRNA was collected at 0 h, 3 h, 6 h, 12 h, 24 h, 36 h, 48 h, and 96 h after *A. hydrophila* or LPS injection. The vertical axis was the relative expression of *MK2*. The analysis of significance was marked with an asterisk. A single asterisk means there is a significant difference (* *p* < 0.05), and double asterisks mean a highly significant difference (** *p* < 0.01).

**Figure 6 genes-13-02060-f006:**
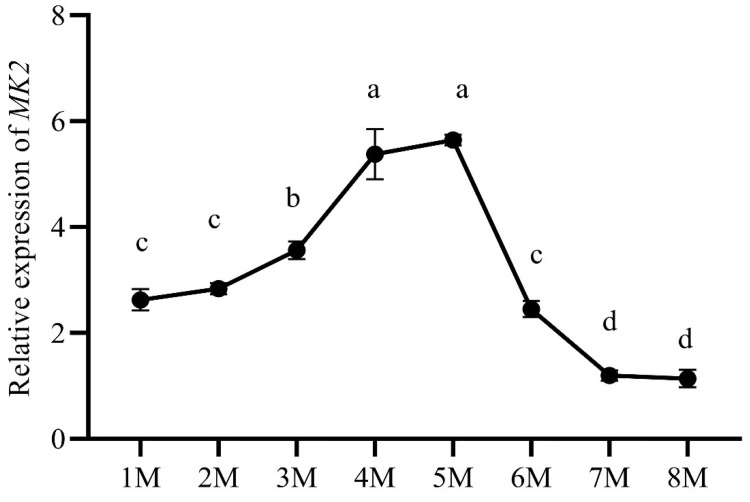
The relative expression of *MK2* during early gonadal development in *H. cumingii*. The horizontal axis represents the gonadal development period (M: months). The vertical axis represents the expression of *MK2*. Different letters (a–d) indicate significant differences (*p* < 0.05). The same letters indicate that there was no significant difference.

**Figure 7 genes-13-02060-f007:**
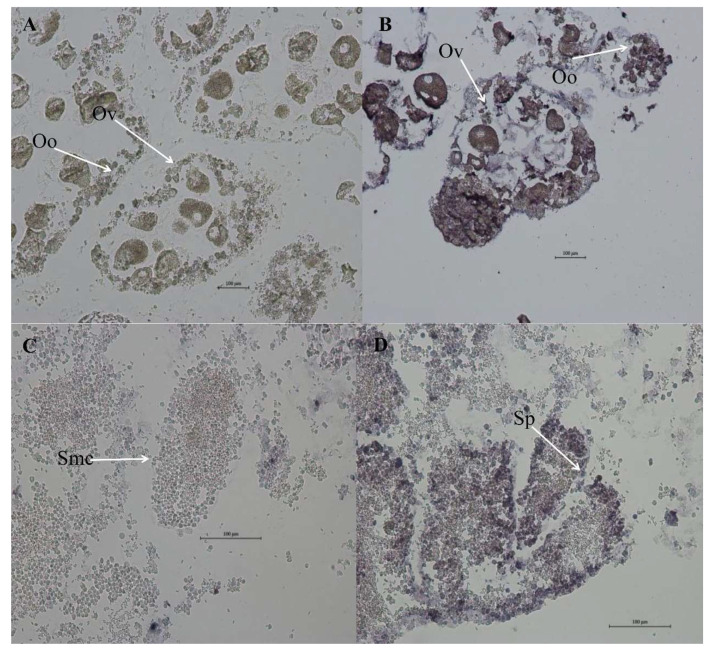
*MK2* in situ hybridization in 1-year-old *H. cumingii* gonads. (**A**) Female-negative control group (**B**) Female experimental group (**C**) Male-negative control group (**D**) Male experimental group (Ov: Ovum; Oo: Oocyte; Smc: Spermatocyte; Sp: Sperm).

**Figure 8 genes-13-02060-f008:**
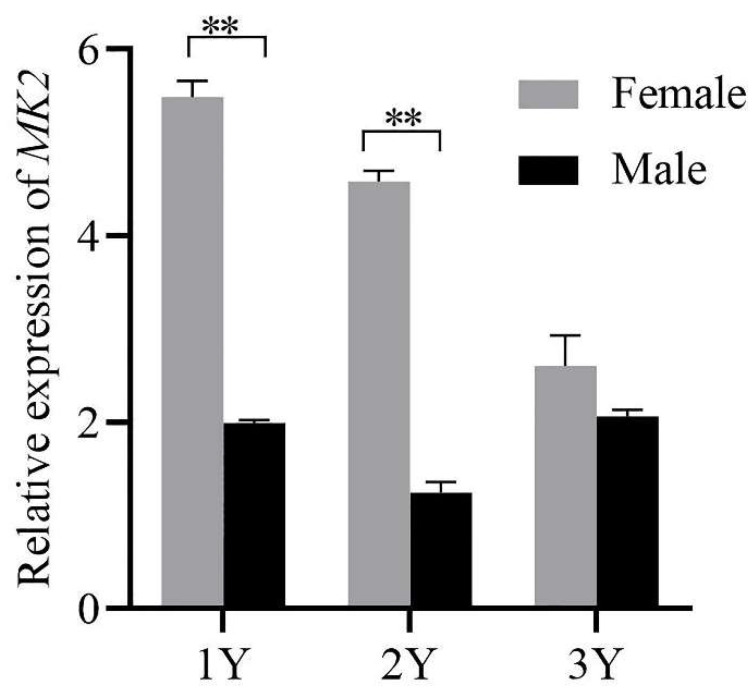
Relative expression of *MK2* in female and male *H. cumingii* aged 1 to 3 years old. *EF1α* was selected as the internal reference gene, and the quantitative results all had three sets of biological replicates. The horizontal axis of the coordinates was the age of the *H. cumingii* (Y: years), and the vertical axis is the relative expression of *MK2*. The analysis of significance was marked with an asterisk. Double asterisks mean a highly significant difference (** *p* < 0.01).

**Table 1 genes-13-02060-t001:** Primers used in this study.

Primer Name	Sequences (5′—3′)	Usage
MK2-Outer	AGAGGTGGCGAAACCCGACAGGACT	3′RACE outer
MK2-Inner	GGAACCGAGTCTCCAAGGATGCCA	3′RACE inner
qPCRMK2-F	CTCGCTAAAATCATTACAGACCC	qPCR
qPCRMK2-R	GAATGGAGGATAGCCACATAACA	qPCR
EFl-αF	GGAACTTCCCAGGCAGACTGTGC	qPCR
EFl-αR	TCAAAACGGGCCGCAGAGAAT	qPCR
IMK2-F	CTCGCTAAAATCATTACAGACCC	ISH
IMK2-R	TAATACGACTCACTATAGGGGAATGG AGGATAGCCACATAACA	ISH

## Data Availability

All experimental data used for supporting the conclusions in this article are reasonably available through the authors.

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
