# Peer review of "Identification and Functional Analysis of MAPKAPK2 in Hyriopsis cumingii"

_genes, 2022, doi:10.3390/genes13112060_

Round 1
Reviewer 1 Report
In this manuscript, MAPKAPK2 (MK2) was cloned and identified from Hyriopsis cumingii. The authors concluded that MK2 might be involved in the immune response and the development of the gonads in H. cumingii. Overall, it can be a valuable contribution to the field. However, several points require attention and should be addressed as described below.
Major issues
1. In Figure 5, control groups injected with solvent are missed. It is not appropriate to compare the expression level between 96h and 0h.
Minor issues
1. The full name of MK2 should be included in the title.
2. It shows "923 bp in 3' UTR" in abstract but "398 bp in 3' UTR" in section 3.1. Please correct.
3. Please modify table 1 following three-line table format.
4. For sectin 2.5. Quantitative real-time PCR, how to calculate the gene expression level and how to analyze the data?
5. Can the authors provide Genbank accession number for all genes or proteins used in this manuscript?
6. In section 2.3. RNA extraction and cDNA synthesis, connected and Transfect should be replaced with cloned and Transform, respectively. Aslo, please use passive voice. For example, The ligation products were transformed to DH5α (TaKaRa, Japan) for sequencing. Check all.
7. Please search the whole text for some minor errors to be corrected.
For example,
In Abstract, (p38MAPK) pathway and 87 bp in 5 'UTR should be replaced with (p38 MAPK) and 87 bp in 5' UTR, respectively.
In Discussion, correct the following sentence.
After direct infection by pathogenic bacteria, will MK2 promote an innate immune response in H. cum-ingii? LPS is a component of the cell wall of Gram-negative bacteria and can be used as immune enhancer for non-specific immune aquatic animals. Additionally, A. hydrophila is the delta mussel breeding culture The main pathogenic bacteria in the breeding process of the mussels.
Author Response
Point 1: In Figure 5, control groups injected with solvent are missed. It is not appropriate to compare the expression level between 96h and 0h.
Response 1: Thanks for your suggestions. In Figure 5, samples at 0h were not stimulated by the pathogen. 0h was used as the control group to detect the change in transcription levels of MK2with time after pathogenic stimulation.
Point 2: The full name of MK2 should be included in the title.
Response 2: Thanks for your suggestions. The full name of MK2 has been added in the title.
Point 3: It shows "923 bp in 3' UTR" in abstract but "398 bp in 3' UTR" in section 3.1. Please correct.
Response 3: Thanks for your suggestions. It was written incorrectly in abstract and have already been coreected.
Point 4: Please modify table 1 following three-line table format.
Response 4: Thanks for your suggestions. Before submission, we refer to an article published in journal Genes in 2022. The tabular form from the reference article was what I showed in my original manuscript. It was used in our origal manuscript. Now, Table 1 has been modified to three-line table format.
Point 5: For sectin 2.5. Quantitative real-time PCR, how to calculate the gene expression level and how to analyze the data?
Response 5: Thanks for your suggestions. It has been provided and revised correspondingly.
Point 6: Can the authors provide Genbank accession number for all genes or proteins used in this manuscript?
Response 6: Thanks for your suggestions. The amino acid accession numbers have been provided correspondingly.
Point 7: In section 2.3. RNA extraction and cDNA synthesis, connected and Transfect should be replaced with cloned and Transform, respectively. Aslo, please use passive voice. For example, The ligation products were transformed to DH5α (TaKaRa, Japan) for sequencing. Check all.
Response 7: Thanks for your suggestions. Incorrect statements have been altered correspondingly.
Point 8: Please search the whole text for some minor errors to be corrected.
Response 8: Thanks for your suggestions. Errors have been corrected and marked up using the “Track Changes” function.
Point 9: In Discussion, correct the following sentence.
After direct infection by pathogenic bacteria, will MK2 promote an innate immune response in H. cumingii? LPS is a component of the cell wall of Gram-negative bacteria and can be used as immune enhancer for non-specific immune aquatic animals. Additionally, A. hydrophila is the delta mussel breeding culture The main pathogenic bacteria in the breeding process of the mussels.
Response 9: Thanks for your suggestions. The sentence indicated in discussion has been corrected.

Reviewer 2 Report
Abstract: if authors are giving full form of ORF, then they should give full form of UTR too. However, both full forms are not necessary. Also, it is good idea to write the full gene name.
Introduction: "Meanwhile, the pearl production performance of male mussels is superior to that of female mussels [21]. Thus, achieving mono-sexual culture or improving breeding capacity will drive the development of the pearl culture industry."..... here, the emphasis is given to achieve mono-sexual culture but the gene of interest is not showing enough expression difference in gonads but it gills. Why is that?
"The thesis aims to explore the role of MK2 in H. cumingii, and provide a scientific and theoretical basis to promote the efficient and sustainable development of H. cumingii culture.".... this is a journal article not thesis.
Materials and methods: no comments
Results: I like the way you represent results. However is there a reason not to perform RNA-Seq analysis?
Discussion and conclusion: I did not find an explanation about how these results will help to develop a mono-sexual pearl culture. If you manage to develop a mono-sexual culture then will that affect environmental diversity?
Author Response
Point 1: Abstract: if authors are giving full form of ORF, then they should give full form of UTR too. However, both full forms are not necessary. Also, it is good idea to write the full gene name.
Response 1: Thank you for your suggestions. We have revised it correspondingly. The full gene name of MK2 has been added in the title.
Point 2: Introduction: "Meanwhile, the pearl production performance of male mussels is superior to that of female mussels [21]. Thus, achieving mono-sexual culture or improving breeding capacity will drive the development of the pearl culture industry."..... here, the emphasis is given to achieve mono-sexual culture but the gene of interest is not showing enough expression difference in gonads but it gills. Why is that?
Response 2: Thank you for your suggestions. According to the results of our research, MK2 might play an important role in the formation of primitive germ cells and innate immune response after pathogenic attack in H. cumingii. MK2 is a differentially expressed gene of gonads screened from transcriptome database. The expression of MK2 has significant differences between male gonads the female gonads in H. cumingii. Studies have shown that MK2, as a significant regulator, is involved in the meiosis and maturation of oocytes. MK2 is one of the downstream substrate of p38 MAPK, which is commonly seen in inflammatory responses, enhancing the transcription levels of inflammatory cytokines. There have been many cases of MK2 being involved in immune responses in different species. Gills are important organs invloved in immune response in H. cumingii. It may be the reason that the gene of interest is showing enough expression difference in gills.
Point 3: "The thesis aims to explore the role of MK2 in H. cumingii, and provide a scientific and theoretical basis to promote the efficient and sustainable development of H. cumingii culture.".... this is a journal article not thesis.
Response 3: Thank you for your suggestions. Sorry for my issues. We will pay more attention to it.
Point 4: Results: I like the way you represent results. However is there a reason not to perform RNA-Seq analysis?
Response 4: Thank you for your suggestions. MK2 is a differentially expressed gene screened from transcriptome database. Study about transcriptome analysis of the female and male gonads in H. cumingii has been published. Please refer to “Wang YY, Duan SH, Wang GL, Li JL. Integrated mRNA and miRNA expression profile analysis of female and male gonads in Hyriopsis cumingii. Sci Rep. 2021 Jan 12;11(1):665. doi: 10.1038/s41598-020-80264-7.”
Point 5: Discussion and conclusion: I did not find an explanation about how these results will help to develop a mono-sexual pearl culture. If you manage to develop a mono-sexual culture then will that affect environmental diversity?
Response 5: Thank you for your suggestions. Explanations about how these results will help to develop a mono-sexual pearl culture have been added in discussion. H. cumingii is a freshwater pearl mussel. The culture mode of H. cumingii is raft culture in confined water. Meanwhile, shellfish are poorly mobile. Developing a mono-sexual culture controls aims to enhance pearl cultivation ability. It won’t affect environmental diversity.

Round 2
Reviewer 1 Report
Point 1: In Figure 5, control groups injected with solvent are missed. It is not appropriate to compare the expression level between 96h and 0h.
Response 1: Thanks for your suggestions. In Figure 5, samples at 0h were not stimulated by the pathogen. 0h was used as the control group to detect the change in transcription levels of MK2with time after pathogenic stimulation.
The manuscript showed "At the end of the temporary care, they were divided into three groups. One group was set as a blank control, and the remaining two groups were used for A. hydrophila and LPS infestation respectively. The concentration of A. hydrophila was made up to 109 CFU/mL. Each H. cumingii. was injected with a 1 ml syringe with A. hydrophila and LPS (1 mg/ml), 50 μl each, in the adductor, and the injected mussels were returned to the same environment for temporary breeding. "
How did the authors treat blank group? What's the solvent in 50 μl A. hydrophila and LPS (1 mg/ml)? The authors were highly recommended to make up the control groups at each time points (3, 6, 12, 24, 36, 48, 60, 96h) in Fig. 5. It is not reasonable that 0h was used as the control group for other treatment group.
Aslo, please check some errors, such as "LPS" and "Each H. cumingii.".
Author Response
Point 1: How did the authors treat blank group? What's the solvent in 50 μl A. hydrophila and LPS (1 mg/ml)? The authors were highly recommended to make up the control groups at each time points (3, 6, 12, 24, 36, 48, 60, 96h) in Fig. 5. It is not reasonable that 0h was used as the control group for other treatment group.
Response 1: Thakns for your suggestions. Experiment of section “2.2. Immune response” has been redone. PBS buffer solution was the solvent in 50 μl A. hydrophila and LPS. Inappropriate statements and results have been corrected correspondingly in the manuscript.